# Improving Deep Echo State Network with Neuronal Similarity-Based Iterative Pruning Merging Algorithm

**Qingyu Shen [1,2,3], Hanwen Zhang [1,2,3] and Yao Mao [1,2,3,*]**

1 Key Laboratory of Optical Engineering, Chinese Academy of Sciences, Chengdu 610209, China
2 Institute of Optics and Electronics, Chinese Academy of Sciences, Chengdu 610209, China
3 University of Chinese Academy of Sciences, Beijing 100049, China
* Correspondence: maoyao@ioe.ac.cn

**Abstract:** Recently, a layer-stacked ESN model named deep echo state network (DeepESN) has been established. As an interactional model of a recurrent neural network and deep neural network, investigations of DeepESN are of significant importance in both areas. Optimizing the structure of neural networks remains a common task in artificial neural networks, and the question of how many neurons should be used in each layer of DeepESN must be stressed. In this paper, our aim is to solve the problem of choosing the optimized size of DeepESN. Inspired by the sensitive iterative pruning algorithm, a neuronal similarity-based iterative pruning merging algorithm (NS-IPMA) is proposed to iteratively prune or merge the most similar neurons in DeepESN. Two chaotic time series prediction tasks are applied to demonstrate the effectiveness of NS-IPMA. The results show that the DeepESN pruned by NS-IPMA outperforms the unpruned DeepESN with the same network size, and NS-IPMA is a feasible and superior approach to improving the generalization performance of DeepESN. The newly proposed method has broad application prospects in real-time systems.

**Keywords:** reservoir computing; deep echo state network; neuronal similarity-based iterative pruning merging algorithm; chaotic time series forecast

## 1. Introduction

Recurrent neural networks (RNNs) represent a consolidated computational abstraction for learning with variable-length time series data [1]. As a simplified paradigm of RNN, the echo state network [2,3] (ESN) provides a prominent reduction in the computational cost compared to other paradigms of RNNs (e.g., [4] (LSTM), [5] (GRU)), for which the hidden layer of ESN is constructed by a randomly generated reservoir instead of independent neurons and the output weights of ESN are trained by a simple linear regression rather than a backpropagation algorithm. Thus, ESN has a successful application in various time series prediction problems (e.g., [6–9]).

Deep neural networks [10] (DNNs) have the potential to learn data representations at various levels of abstraction and are being increasingly stressed in the machine learning community. Recently, a layer-stacked ESN model named deep echo state network (Deep-ESN) was established and investigated, theoretically and experimentally, by Gallicchio et al: the inherent characterization of the system dynamics developed at the different layers of DeepESN is experimentally analyzed in [11] and theoretically explained in [12]; A theoretical foundation for the study of DeepESN from a dynamical system point of view is introduced in [13]; further details on the analysis and advancements of DeepESN could be found in [14]. As an interactional model of RNN and DNN, investigations of DeepESN are of significant importance in both areas. On the one hand, DeepESN expands our knowledge on how the information with memory attracted by RNN is extracted by hierarchical neural networks; on the other hand, DeepESN helps us better understand how the abstract intrinsic representations of time series extracted by DNN are recalled

in reservoirs. Furthermore, DeepESN has richer nonlinear representation capacity, less computational complexity, and better predictive performance than single-layer ESN [15].

Optimizing the structure of neural networks remains a common task in artificial neural networks, and the same question of how many neurons should be used in each layer must be stressed in all types of neural networks. If the neurons are too few, the architecture does not satisfy the error demand by learning from the data, whereas if there are too many neurons, learning leads to the well-known overfitting problem [16]. As far as we know, minimal research has been carried out on optimizing the architecture of DeepESN. For research that has already been carried out on DeepESN, the same number is commonly assigned in each layer, which is acceptable but not optimal. In this paper, our aim is to solve the problem of choosing the optimized size of DeepESN, especially the number of neurons in different layers.

In 2014, a sensitive iterative pruning algorithm (SIPA) was proposed by Wang and Yan [17] to optimize the simple cycle reservoir network (SCRN). The algorithm was used to prune the least sensitive neurons one by one according to the sensitivity analysis, and the results showed that the SIPA method can optimize the structure and improve the generalization performance of the SCRN; meanwhile, pruning out redundant neurons could contribute to reducing the calculation and improving the computing efficiency of the network. Inspired by these advantages of SIPA, we wanted to apply a similar iterative pruning approach to DeepESN. However, SCRN is a kind of minimal-complexity ESN with a simple cycle topology, and the topology of DeepESN is much more complex than SCRN. Pruning a neuron in the network will raise perturbations on adjacent neurons, resulting in unstable network performance; in SIPA, the perturbations could be eliminated by adjusting the input weights into the perturbed neurons to minimize the distance of its input signal before and after pruning. Due to the hierarchical structure of reservoirs in DeepESN, perturbation raised by pruning a neuron in the lower layer will propagate into higher layers layer-by-layer, leading to greater instability of network performance and the difficulty of perturbation elimination.

In order to overcome the above difficulty, a new neuronal similarity-based iterative pruning merging algorithm (NS-IPMA) is proposed to iteratively prune out or merge the most similar neurons in DeepESN. In NS-IPMA, a pair of the most similar twin neurons, which are regarded as redundant neurons in the network, are selected out iteratively; then, if they exist in different layers, the one in the higher layer will be pruned out, and if they are in the same reservoir, they will be merged into one neuron, which works as the substitution of antecedent twin neurons. Quantitative estimation of neuronal similarity plays an essential role in determining the redundant neurons that should be pruned out. Four neuronal similarity estimation criteria of NS-IPMA approaches were attempted, including the inverse of Euclidean distance, Pearson's correlation, Spearman's correlation and Kendall's correlation. Reducing the network size is a directly effective approach to improve the generalization performance of a neural network because pruning out neurons will lead to a reduction in network size to verify the effectiveness of the NS-IPMA method. The DeepESNs pruned by the NS-IPMA method were compared with unpruned DeepESNs, whose number of neurons in each reservoir is the same. The pruned DeepESN and the unpruned DeepESN were compared with an equal number of layers and equal number of total neurons. The results of the experiment on two chaotic time series prediction tasks showed that the NS-IPMA method has good network structure adaptability, and the DeepESNs pruned by the NS-IPMA method has better generalization performance and better robustness than the unpruned DeepESNs, indicating that the NS-IPMA method is a feasible and superior approach to improving the generalization performance of DeepESN. The NS-IPMA method provides a novel approach for choosing the appropriate network size of DeepESN, and it also has application potential in other RNNs and DNNs.

Our highlight contributions consist of:

- A new NS-IPMA that works effectively in improving the generalization performance of DeepESN is proposed.
- The proposed method contributes to simplifying the structure of DeepESN and reducing computational cost.

This paper is organized as follows: DeepESN and entropy quantification of reservoir richness are introduced in detail in Section 2, SIPA and the newly proposed NS-IPMA are described in Section 3, experiments and results are presented and discussed in Sections 4 and 5, and Section 6 draws conclusions and highlights the future direction.

## 2. Deep Echo State Network

### 2.1. Leaky Integrator Echo State Network

A leaky integrator echo state network [18], (LI-ESN), as shown in Figure 1, is a recurrent neural network with three layers: input layer $\mathbf{u}(t) \in \mathbb{R}^{N_u \times 1}$, hidden layer $\mathbf{x}(t) \in \mathbb{R}^{N_x \times 1}$, output layer $\mathbf{y}(t) \in \mathbb{R}^{N_y \times 1}$. $t$ denotes time sequence order. The hidden layer is regarded as a reservoir, which holds the memory of foregone information, and $\mathbf{x}(t)$ is refreshed by the state transition function:

$$\mathbf{x}(t) = \alpha \mathbf{x}(t-1) + (1-\alpha)\tanh(\mathbf{W}_i \mathbf{u}(t-1) + \mathbf{W}_r \mathbf{x}(t-1)), \qquad (1)$$

where $\mathbf{W}_i \in \mathbb{R}^{N_x \times N_u}$ is the input weight matrix randomly generated before training, and $\mathbf{W}_r \in \mathbb{R}^{N_x \times N_x}$ is the reservoir weight matrix previously given before training. $\alpha \in [0, 1]$ is the leaky parameter. $\tanh(\bullet)$ is the activation function of the hidden layer. The reservoir weights in $\mathbf{W}_r$ must be initialized to satisfy the echo state property (ESP) [19,20], denoting by $\rho(\bullet)$ the spectral radius operator (i.e., the largest absolute eigenvalue of its matrix argument), and the necessary condition for the ESP is expressed as follows:

$$\rho((1-\alpha)\mathbf{I} + \alpha \mathbf{W}_r) < 1. \qquad (2)$$

Accordingly, the values in matrix $\mathbf{W}_r$ are randomly selected from a uniform distribution (e.g., $U[-1, 1]$), and then rescaled to satisfy the above condition in Equation (2).

reservoir

**Figure 1.** Structure of LI-ESN.

The output $\mathbf{y}(t)$ can be calculated through a linear combination of reservoir states as follows:

$$\mathbf{y}(t) = \mathbf{W}_o \mathbf{x}(t), \qquad (3)$$

where $\mathbf{W}_o \in \mathbb{R}^{N_y \times N_x}$ is the output weight matrix.

During training, the states of reservoir neurons are collected in a training state matrix $\mathbf{X}_{train} = [\mathbf{x}(1), \mathbf{x}(2) \ldots \mathbf{x}(L_{train})]$, and an output target matrix $\mathbf{Y}_{train} = [\hat{\mathbf{y}}(1), \hat{\mathbf{y}}(2) \ldots \hat{\mathbf{y}}(L_{train})]$ is collected correspondingly, where $L_{train}$ is the number of training samples.

The output weights in $\mathbf{W}_o$ can be calculated by ridge regression as follows:

$$\mathbf{W}_o = \left( (\mathbf{X}_{train}^T \mathbf{X}_{train} + \lambda \mathbf{I})^{-1} \mathbf{X}_{train}^T \mathbf{Y}_{train} \right)^T, \tag{4}$$

where $(\bullet)^T$ represents matrix transpose, $(\bullet)^{-1}$ represents matrix inversion, $\lambda \mathbf{I}$ is a regularization term ensuring $\mathbf{X}_{train}^T \mathbf{X}_{train}$ is invertible.

### 2.2. Deep Echo State Network

DeepESN, as shown in Figure 2, was first introduced by Gallicchio [11,14], as a stacked reservoir computing (RC) architecture, where multiple reservoir layers are stacked on top of each other. The state transition functions of hidden layers in DeepESN are expressed as:

$$\begin{cases} \mathbf{x}^1(t) = \alpha^1 \mathbf{x}^1(t) + (1 - \alpha^1) \tanh(\mathbf{W}_i \mathbf{u}(t-1) + \mathbf{W}_r^1 \mathbf{x}^1(t-1)), & l = 1 \\ \mathbf{x}^l(t) = \alpha^l \mathbf{x}^l(t) + (1 - \alpha^l) \tanh(\mathbf{W}_p^{l-1} \mathbf{x}^{l-1}(t-1) + \mathbf{W}_r^l \mathbf{x}^l(t-1)), & l \in [2, L] \end{cases} \tag{5}$$

where the superscript(1 and $l$) is the layer notation, with a total of $L$ hidden layers in the network. $\mathbf{x}^l(t) \in \mathbb{R}^{N_x^l \times 1}$ represents the l-th hidden layer (i.e., reservoir($l$)) with $N_x^l$ neurons inside, $\mathbf{W}_i \in \mathbb{R}^{N_x^1 \times N_u}$ is the input weight matrix of the first hidden layer, $\mathbf{W}_r^l \in \mathbb{R}^{N_x^l \times N_x^l}$ is the reservoir weight matrix of the $l$-th hidden layer, $\mathbf{W}_p^{l-1} \in \mathbb{R}^{N_x^{l-1} \times N_x^l}$ is the propagate weight matrix, which connects reservoir($l-1$) to reservoir($l$).

As in the standard LI-ESN approach, the reservoir weights of a DeepESN are initialized subject to similar stability constraints. In the case of DeepESN, such constraints are expressed by the necessary conditions for the ESP of deep RC networks [13] described by the following equation:

$$\max_{l \in [1,L]} \rho \left( (1 - \alpha^l) \mathbf{I} + \alpha^l \mathbf{W}_r \right) < 1, \tag{6}$$

where the same leaky parameter ($\alpha^l \equiv \alpha$) in each layer is considered in this paper.

The values in each reservoir matrix $\{\mathbf{W}_r^l | l \in [1, L]\}$ are randomly initialized from a uniform distribution (e.g., $U[-1, 1]$); after that, each $\mathbf{W}_r^l$ is spectrally normalized by its spectral radius and rescaled by the same reservoir scaling parameter $\gamma_r$ to meet the demands of Equation (6).

$$\mathbf{W}_r^l \leftarrow \frac{\gamma_r \mathbf{W}_r^l}{\rho(\mathbf{W}_r^l)}, \quad l \in [1, L]. \tag{7}$$

The values in input weight matrix $\mathbf{W}_i$ are randomly selected from a uniform distribution $U[-\gamma_i, \gamma_i]$, where $\gamma_i$ is the input scaling parameter. The values in each propagate weight matrix $\{\mathbf{W}_p^l | l \in [1, L-1]\}$ are randomly selected from a uniform distribution $U[-\gamma_p, \gamma_p]$, where $\gamma_p$ is the propagate scaling parameter.

The output equation and the training equation of DeepESN are formed by concatenating all hidden neurons in all reservoirs together, denoting $\tilde{\mathbf{x}}(t)^T = [\mathbf{x}^1(t)^T \ \mathbf{x}^2(t)^T \dots \mathbf{x}^L(t)^T]$ and substituting $\tilde{\mathbf{x}}(t)$ for $\mathbf{x}(t)$ in Equations (3) and (4):

$$\mathbf{y}(t) = \mathbf{W}_o \tilde{\mathbf{x}}(t), \tag{8}$$

$$\mathbf{W}_o = \left( (\tilde{\mathbf{X}}_{train}^T \tilde{\mathbf{X}}_{train} + \lambda \mathbf{I})^{-1} \tilde{\mathbf{X}}_{train}^T \mathbf{Y}_{train} \right)^T. \tag{9}$$

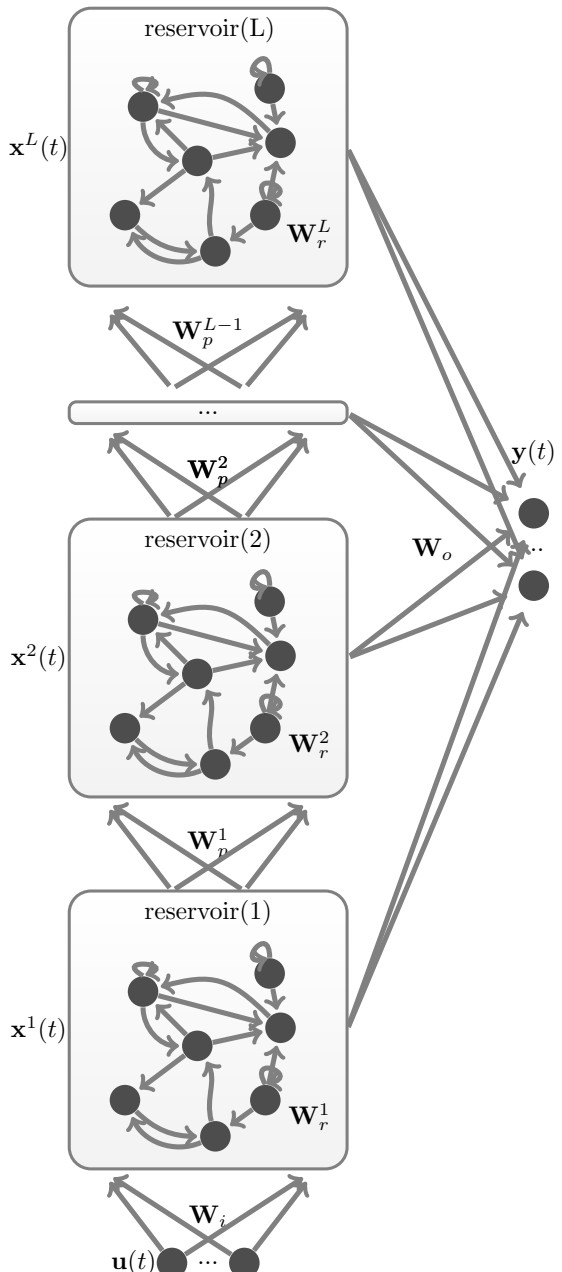

**Figure 2.** Structure of DeepESN.

### 2.3. Architectural Richness of DeepESN

The components of the state should be as diverse as possible to provide a richer pool of dynamics from which the trainable part can appropriately combine. From an information-theoretic point of view, this form of richness could be measured by means of the entropy of instantaneous reservoir states [1]. Here, an efficient estimator of Renyi's quadratic entropy is introduced: suppose that we have N independent and identically distributed samples $\{v_1, \dots, v_N\}$ for the continuous random variable **V**. An estimation of Renyi's quadratic entropy directly from the sampling data is defined as:

$$H_2(\mathbf{V}) = -\log\left(\frac{1}{N^2}\sum_{i=1}^{N}\sum_{j=1}^{N}G_{\kappa\sqrt{2}}(v_j - v_i)\right), \tag{10}$$

where $G_{\kappa\sqrt{2}}(\bullet)$ is a Gauss kernel function with standard deviation $\kappa\sqrt{2}$, and $\kappa$ can be determined by Silverman's rule:

$$\kappa = \sigma(\mathbf{V})(4N^{-1}(2d+1)^{-1})^{\frac{1}{d+4}}, \tag{11}$$

where $\sigma(\mathbf{V})$ is the standard deviation, and $d$ is the data dimensionality.

Average state entropy (ASE) is obtained by time averaging the instantaneous Renyi's quadratic entropy estimation of reservoir neurons.

$$\mathcal{H}(t) = H_2(\tilde{\mathbf{x}}(t)), \tag{12}$$

$$ASE = \frac{1}{S}\sum_{t=1}^{S}\mathcal{H}(t), \tag{13}$$

where $S$ is the sample number. ASE gives us a research perspective independent of the learning aspect, and higher ASE values are preferable and denote richer dynamics in reservoirs [1].

## 3. Pruning Deep Echo State Network with a Neuronal Similarity-Based Iterative Pruning Merging Algorithm

### 3.1. Sensitive Iterative Pruning Algorithm to Simple Cycle Reservoir Network

The SCRN is a kind of minimum complexity ESN, which has a cycle topology in the reservoir [21]. Every reservoir neuron is unidirectionally connected to its two adjacent neurons. An SIPA method is introduced to choose the right network size for SCRN by iteratively pruning out the least sensitive neurons [17].

The SIPA method is carried out in the following steps:

1.  Establish an SCRN with a large enough reservoir and satisfactory performance.
2.  Select a neuron to be pruned using the sensitive criterion [22] (assume $x_m$ is to be pruned), as diagramed in Figure 3, and remove all weights connected to $x_m$ as follows:

$$\begin{cases} \mathbf{W}_i(m,:) \leftarrow \mathbf{0} \\ \mathbf{W}_r(m, m-1) \leftarrow 0 \\ \mathbf{W}_r(m+1, m) \leftarrow 0 \end{cases} \tag{14}$$

3.  Establish a new link between two neighbors of pruned neurons, the link weight is determined to eliminate the perturbation caused by pruning, denoting the input to $x_{m+1}$ before pruning $I_o = \mathbf{W}_i(m+1,:)\mathbf{u} + \mathbf{W}_r(m+1, m)x_m$, the input to $x_{m+1}$ after pruning $I_n = \mathbf{W}_i(m+1,:)\mathbf{u} + \mathbf{W}_r(m+1, m-1)x_{m-1}$, the perturbation is eliminated, and the original reservoir behavior is maintained as long as $I_n$ is set as close as possible to $I_o$, by solving the following optimization problem:

$$\min_{\mathbf{W}_r(m+1, m-1)} ||I_o - I_n||_2. \tag{15}$$

4.  Adjust the output weights by retraining the network, and then calculate the training error.
5.  Repeat steps 2–4 until the training error or the reservoir size reaches an acceptable range.

The key to a successful application of SIPA is Step 3. In Step 2, a reservoir perturbation is triggered by pruning, making the performance of the pruned network unpredictable. Thus, the essential task of Step 3 is to reduce the effects of perturbation so that the rest of the neurons remain unchanged and the network performance is approximately as good as before.

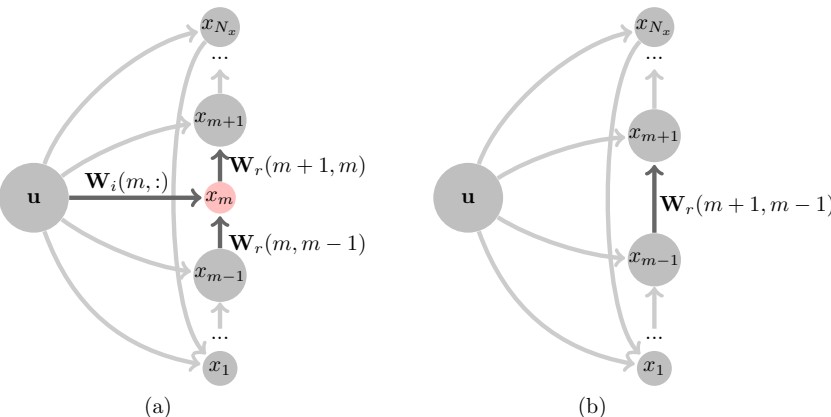

**Figure 3.** Weight coefficient diagram of connections of neurons in the reservoir of SCRN before (**a**) and after (**b**) pruning a neuron by SIPA.

### 3.2. Neuronal Similarity-Based Iterative Pruning Merging Algorithm to Deep Echo State Network

Due to the simple topology of SCRN, only one neuron ($x_{m+1}$) receives input from the pruned neuron ($x_m$), and perturbation elimination in SCRN is easy to perform. However, the topology of DeepESN is much more complicated, and more extensive perturbation will be raised by pruning one neuron in DeepESN because, in a highly coupled reservoir, perturbation at any neuron will be diffused to every neuron of the same reservoir. Furthermore, in hierarchically stacked reservoirs, perturbation in the lower layer will be transmitted to every higher layer above, and it is very difficult to eliminate all these perturbations.

The merging method is designed to solve this difficulty, and this coincides with Islam's idea [23], where two neurons are merged by averaging their input weights. Consider the following ideal scenario: merging two identical twin neurons in the same reservoir will derive a new neuron that is identical to two antecedent twins; this new burned neuron could act as an equivalent substitution for two antecedent twins. Consequently, a neuron is pruned without leading any perturbation through superimposing the output weights of the merged twins as well. This ideal perturbation-free merging has the prerequisite that two identical neurons have to be found in the same reservoir. The more similar the two merged neurons are, the weaker perturbation that will be raised by merging. Neuronal similarity could be assessed by some quantitative relations of the collected training state matrix. Distance and correlation are commonly used to quantify similarity, and four similarity estimation criteria are given in this paper, including the inverse of the Euclidean distance (*ED*), Pearson's correlation coefficient (*PC*), Spearman's correlation coefficient (*SC*) and Kendall's correlation coefficient (*KC*) as follows:

Noting the total number of neurons in all reservoirs $M = \sum_{l=0}^{L} N_x^l$, recall the train state matrix

$$\tilde{\mathbf{X}}_{train} = \begin{bmatrix} \mathbf{x}^1(1) & \mathbf{x}^1(2) & \dots & \mathbf{x}^1(L_{train}) \\ \mathbf{x}^2(1) & \mathbf{x}^2(2) & \dots & \mathbf{x}^1(L_{train}) \\ \vdots & \vdots & \ddots & \vdots \\ \mathbf{x}^L(1) & \mathbf{x}^L(2) & \dots & \mathbf{x}^L(L_{train}) \end{bmatrix}_{(M \times L_{train})} = \begin{bmatrix} \mathbf{n}_1 \\ \mathbf{n}_2 \\ \vdots \\ \mathbf{n}_M \end{bmatrix}, \quad (16)$$

where $L_{train}$ is the number of training samples, and $\mathbf{n}_i$ is the historical state of the i-th neuron state during training. The similarity of $\mathbf{n}_i$ and $\mathbf{n}_j$ is derived by:

$$ED(i,j) = \frac{1}{\| \mathbf{n}_i - \mathbf{n}_j \|_2}, \tag{17}$$

$$PC(i,j) = \frac{\sigma_{\mathbf{n}_i\mathbf{n}_j}}{\sqrt{\sigma_{\mathbf{n}_i\mathbf{n}_i}\sigma_{\mathbf{n}_j\mathbf{n}_j}}}, \tag{18}$$

$$SC(i,j) = 1 - \frac{6\sum_{t=1}^{L_{train}} d_t^2}{L_{train}\left(L_{train}^2 - 1\right)}, \tag{19}$$

$$KC(i,j) = \frac{c - d}{\frac{1}{2}L_{train}(L_{train} - 1)}, \tag{20}$$

where $\sigma_{\mathbf{n}_i\mathbf{n}_j}$ represents the cross-correlation of $\mathbf{n}_i$ and $\mathbf{n}_j$, and $\sigma_{\mathbf{n}_i\mathbf{n}_i}$ and $\sigma_{\mathbf{n}_j\mathbf{n}_j}$ represent the autocorrelation of $\mathbf{n}_i$ and $\mathbf{n}_j$. $d_t$ is the rank difference of $\mathbf{n}_i(t)$ and $\mathbf{n}_j(t)$, $c$ is the number of concordant pairs, and $d$ is the number of discordant pairs in $\mathbf{n}_i$ and $\mathbf{n}_j$. NS-IPMA based on different similarity estimation criteria are named correspondingly; for instance, ES-IPMA means NS-IPMA based on the inverse of Euclidean distance criterion, etc. The main process of NS-IPMA is illustrated in Figure 4, and the NS-IPMA method is carried out in the following steps:

1. Initially generate a performable DeepESN with large enough reservoirs by tuning hyperparameters to minimize the average of training and validate error using the particle swarm optimization (PSO) algorithm; please refer to Appendix A for more details on hyperparameter tuning. This DeepESN is a primitive network to implement.
2. Washout the reservoirs, and activate the reservoirs using training samples to obtain the training state matrix.
3. Quantify the similarity of every two neurons (using one criterion of Equations (17)–(20)), and select a pair of the most similar neurons.
4. (a) If selected neurons are in the same reservoir (note as $x_i^l$ and $x_j^l$), merge them. As diagramed in Figure 5, $x_s^l$ is the son neurons merged as the substitute of its parents $x_i^l$ and $x_j^l$. $l$ is the reservoir layer where $x_i^l$ and $x_j^l$ exist. The related weight matrix $\mathbf{W}_p^{l-1}$, $\mathbf{W}_p^l$ and $\mathbf{W}_r^l$ is refreshed as follows:

$$\begin{cases} \mathbf{W}_p^{l-1}(s,:) \leftarrow \dfrac{\mathbf{W}_p^{l-1}(i,:) + \mathbf{W}_p^{l-1}(j,:)}{2} \\ \mathbf{W}_r^l(s,:) \leftarrow \dfrac{\mathbf{W}_r^l(i,:) + \mathbf{W}_r^l(j,:)}{2} \\ \mathbf{W}_r^l(:,s) \leftarrow \mathbf{W}_r^l(:,i) + \mathbf{W}_r^l(:,j) \\ \mathbf{W}_p^l(:,s) \leftarrow \mathbf{W}_p^l(:,i) + \mathbf{W}_p^l(:,j) \end{cases} \tag{21}$$

   if $l = 1$, $\mathbf{W}_i$ perform as $\mathbf{W}_p^{l-1}$; if $l = L$, $\mathbf{W}_p^l$ does not exist.

   (b) If selected neurons are in different reservoirs (note as $x_i^m$ and $x_j^l$), prune one in a high layer (assume $m < l$). The related weight matrix $\mathbf{W}_p^{l-1}$, $\mathbf{W}_p^l$ and $\mathbf{W}_r^l$ is refreshed as follows:

$$\begin{cases} \mathbf{W}_p^{l-1}(j,:) \leftarrow \mathbf{0} \\ \mathbf{W}_r^l(j,:) \leftarrow \mathbf{0} \\ \mathbf{W}_r^l(:,j) \leftarrow \mathbf{0} \\ \mathbf{W}_p^l(:,j) \leftarrow \mathbf{0} \end{cases} \tag{22}$$

   if $l = 1$, $\mathbf{W}_i$ perform as $\mathbf{W}_p^{l-1}$; if $l = L$, $\mathbf{W}_p^l$ does not exist.
5. Adjust the output weights by retraining the network, and then estimate the performance of the current network.
6. Repeat steps 2–5 until the training error or the network size reaches an acceptable range.

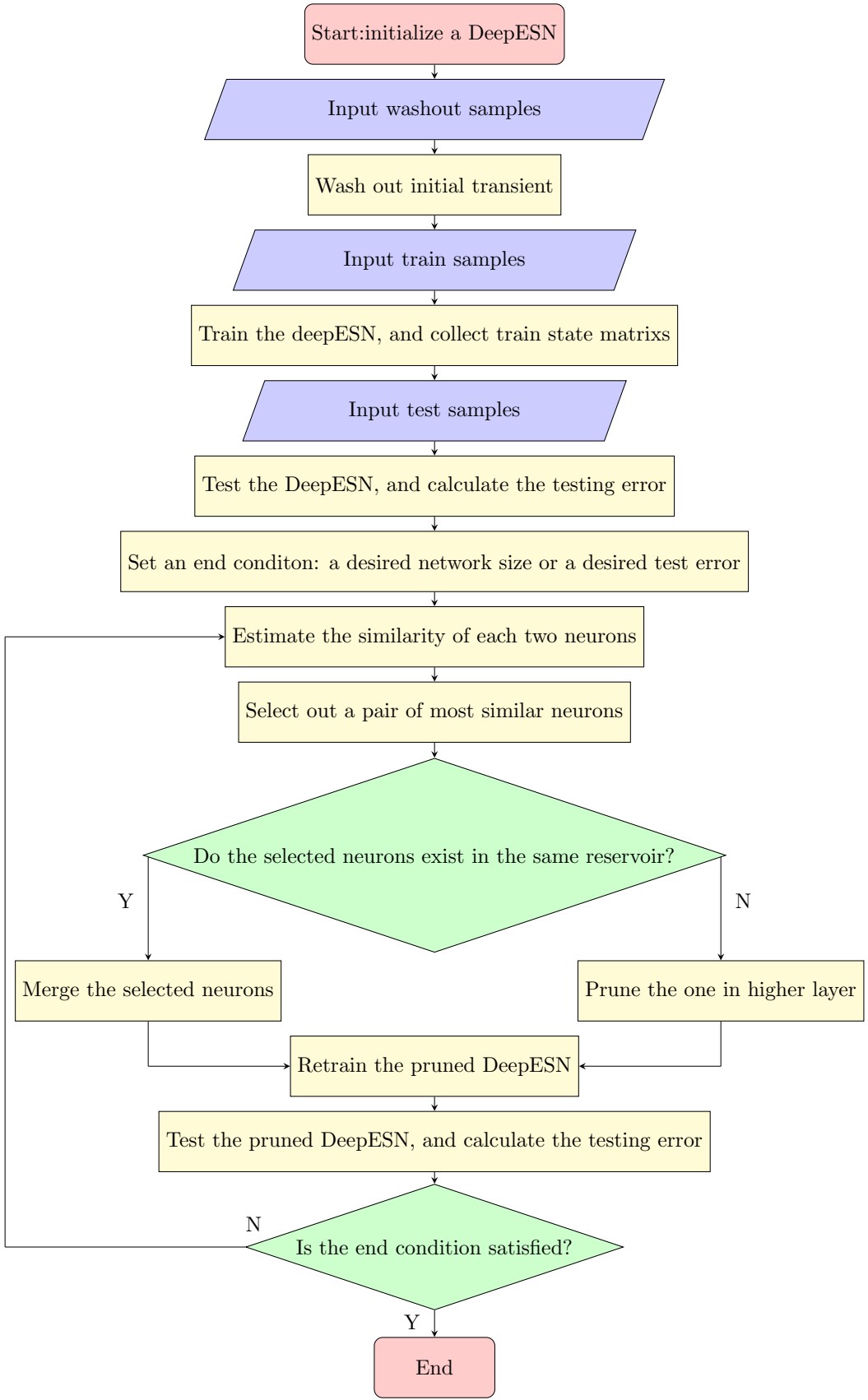

**Figure 4.** Flowchart of NS-IPMA.

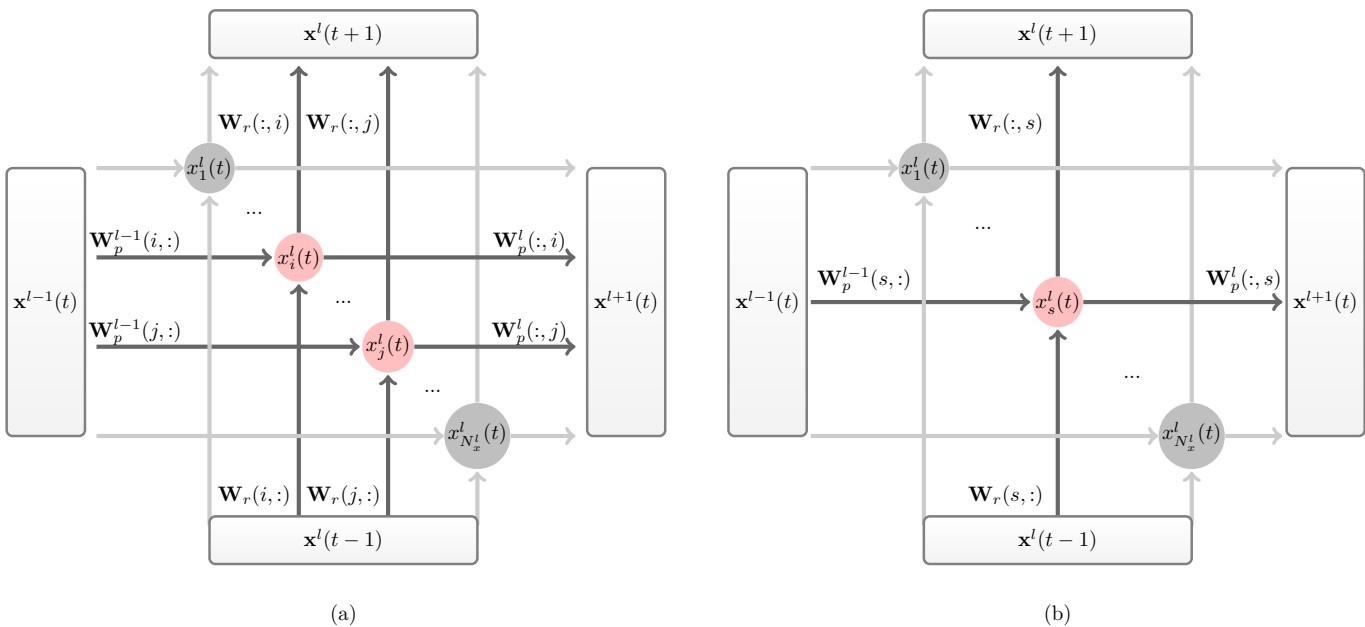

**Figure 5.** Weight coefficients diagram of connected neurons in reservoir (*l*) of DeepESN before (**a**) and after (**b**) merging a neuron by NS-IPMA.

## 4. Experiments

Due to high nonlinearity and instability, forecasting chaotic time series has remained a difficult task for the last few decades [24]. RC architectures, including ESNs, have successfully been employed for multi-step-ahead prediction of nonlinear time series and modeling chaotic dynamical systems at a low computational cost [25]; therefore, chaotic time series prediction is an ideal task to estimate the performance of DeepESN.

### 4.1. Datasets

#### 4.1.1. Mackey–Glass Chaotic Time Series

The Mackey–Glass time series [26] is a standard benchmark for chaotic time series forecast models. The Mackey-Glass time series is defined in the following differential equation:

$$\frac{dx}{dt} = a \frac{x_\tau}{1 + x_\tau{}^c} - bx, \quad x_\tau = x(t - \tau), \tag{23}$$

where $a = 0.1$, $b = 0.2$, and $c = 11$ are constant parameters. The nonlinearity of the system increases as the time delay parameter $\tau$ increases. The system shows chaotic behavior when $\tau \geqslant 17$. To generate the time series used here, $\tau$ is set to 31, and the integration step size is set to $\Delta t = 0.002$ s using the jitcdde Python engine to solve delay differential equations at discrete equally spaced times. The data were then sampled by $4\Delta t$ to form the Mackey–Glass (*MG*) dataset with 4000 data points. A Python library [27] was used to implement this process.

#### 4.1.2. Lorenz Chaotic Time Series

The second chaotic time series benchmark is derived from the Lorenz system [28], which is given by the following equations:

$$\begin{cases} \dfrac{dx}{dt} = -a(x - y), \\ \dfrac{dy}{dt} = -b(x - y) + xz, \\ \dfrac{dz}{dt} = -cz + xy, \end{cases} \tag{24}$$

where $a = \frac{34}{3}$, $b = \frac{298}{11}$, and $c = \frac{17}{7}$ are the constant parameters. The time series is obtained by numerical integration of the equation using odeint, a Python solver of the ordinary differential equation system, where the solution is evaluated at times spaced $\Delta t = 0.08$ s apart, and the initial values are set to $[x(0), y(0), z(0)] = [0.5, 0, -0.5]$, and the Lorenz z-axis (*LZ*) dataset is obtained by taking the z-axis data from 4000 samples.

Before carrying out the prediction task, both datasets are shifted by their mean to remove the DC bias.

### 4.2. Next Spot Prediction Task

In the next spot prediction experiment, the last four continuous spots were considered the input and the next spot was considered the desired output, i.e., $\{u(t-3), u(t-2), u(t-1), u(t)\}$ was used to predict $u(t+1)$. The number of the train samples and the test samples was 1800 and 2000, 200 samples before the first training sample was used to wash out the initial transient (see Figure 6). The reservoir diversity was quantized by the ASE (see: Section 2.3) of the combined training and testing neuron states activated by training and testing samples. The predictive error performance was quantized by the normalized root mean square error:

$$NRMSE = \sqrt{\sum_{t=1}^{T} \frac{\mid \mathbf{y}(t) - \hat{\mathbf{y}}(t) \mid^2}{T \mid \mathbf{y}(t) - \bar{\mathbf{y}}(t) \mid^2}}, \tag{25}$$

where $T$ is sample number, $\hat{\mathbf{y}}(t)$ is the desired output, $\mathbf{y}(t)$ is the readout output, and $\bar{\mathbf{y}}(t)$ is the average of $\mathbf{y}(t)$.

Two different initial reservoir size conditions of DeepESN were performed on both datasets. The first is four stacked reservoirs with 100 neurons in each reservoir (abbreviated to: $4 \times 100$), and the second is eight stacked reservoirs with 50 neurons in each reservoir (abbreviated to: $8 \times 50$). All experiments were carried out under two different initial conditions on two datasets, and model hyperparameters tuned by PSO were recorded in Table A1.

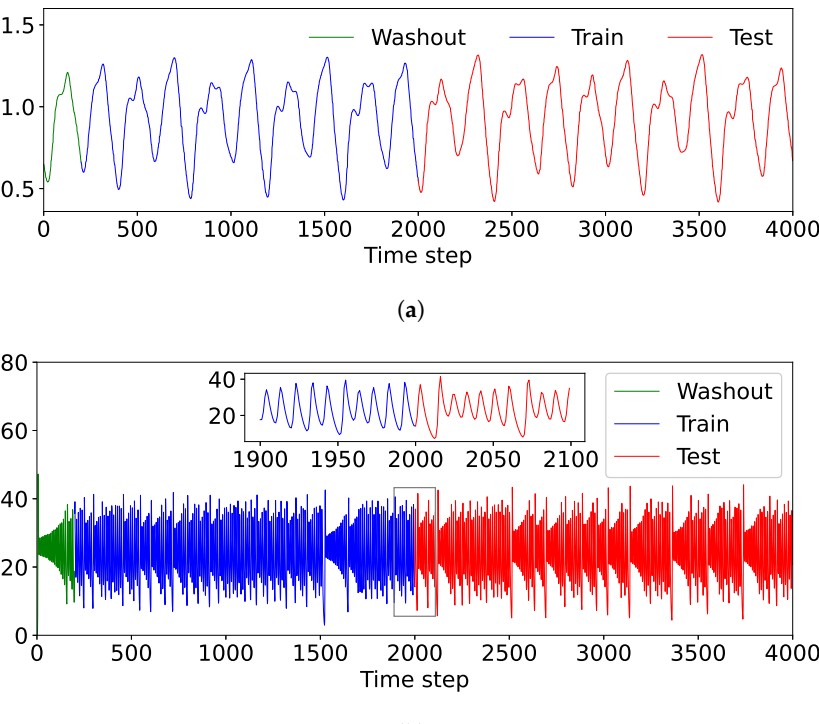

**Figure 6.** *MG* (**a**) and *LZ* (**b**) datasets.

*4.3. Ablation Experiment and Control Experiment*

In order to demonstrate the effectiveness of different similarity estimation criteria, the worst-case scenario of NS-IPMA, the iterative pruning merging algorithm (IPMA) without neuronal similarity estimation, was investigated as an ablation experiment. In IPMA, the similarity of each pair of two neurons was assigned by random values; thus, two random neurons would be recognized as the most similar pair of neurons and would be pruned (or merged).

To verify the effectiveness of the NS-IPMA method, the pruned DeepESNs were compared with a control experiment. The unpruned DeepESN, whose number of neurons in each reservoir is the same, the pruned DeepESN, and the unpruned DeepESN were compared with an equal number of layers, equal number of total neurons and the same hyperparameters because the unpruned DeepESN is a standard benchmark that identifies the evolutionary characteristic of network performance by reducing network size. Ninety percent of neurons of randomly initialized DeepESNs were continuously pruned using different criterion-based NS-IPMA methods (ED-IPMA, PC-IPMA, SC-IPMA, KC-IPMA) and non-criterion-based IPMA. During pruning, networks were silhouetted, and the performance was evaluated once 10% of neurons had been pruned. These pruned groups were compared with unpruned DeepESN. All experiments were repeated 20 times, and all results were averaged through 20 independent replications.

## 5. Results and Discussion

*5.1. Hierarchical Structure*

The number of neurons remaining during pruning in each reservoir of DeepESN, which was pruned by different similarity estimation criterion-based NS-IPMA methods, is shown in Figure 7. As NS-IPMA goes on, we observed a significant reduction in the neuron number in high layers at a later stage. The reason for this was the discard policy, in which when a redundant pair of the most similar neurons were found in different reservoirs, the one in the higher layer would be pruned out. Lower layer reservoirs are the foundation of higher layer reservoirs, and too few neurons in lower layers bring the risk of insufficient information presentation and extraction in higher layers; thus, the NS-IPMA methods have good network structure adaptability.

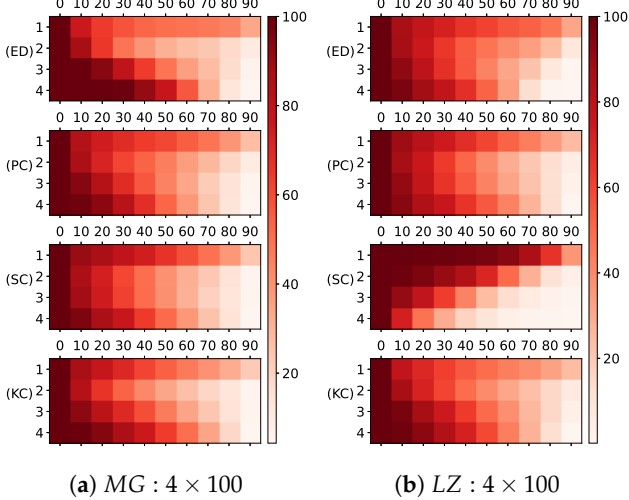

(**a**) $MG : 4 \times 100$       (**b**) $LZ : 4 \times 100$

**Figure 7.** *Cont.*

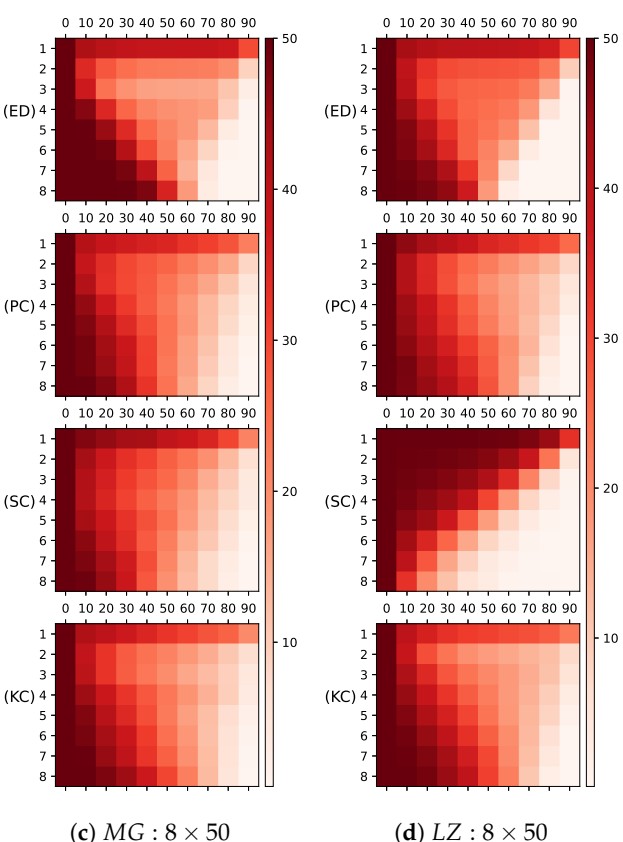

**(c)** *MG* : 8 × 50          **(d)** *LZ* : 8 × 50

**Figure 7.** The figure shows the number of neurons remaining in each layer of DeepESN, pruned by different similarity estimation criterion-based NS-IPMA methods. The vertical axis indicates the layer index, the horizontal axis indicates the percent of pruned neurons in the initial total number of neurons, the mesh color indicates the number of remaining neurons, and the darker color indicates more neurons remaining. (**a**): Initial 4 layer reservoirs with 100 neurons in each reservoir on *MG* dataset; (**b**): Initial 8 layer reservoirs with 50 neurons in each reservoir on *MG* dataset; (**c**): Initial 4 layer reservoirs with 100 neurons in each reservoir on *LZ* dataset; (**d**): Initial 8 layer reservoirs with 50 neurons in each reservoir on *LZ* dataset.

### 5.2. Reservoir Diversity and Error Performance

Quantitative performance comparisons (ASE, training NRMSE, and testing NRMSE) of unpruned DeepESN and DeepESN, which were pruned by IPMA and different similarity estimation criterion-based NS-IPMA methods, are illustrated in Figure 8–10. The results on the two datasets were similar but different. On the whole, as reservoir size reduced, the ASE would decrease and training NRMSE would increase, confirming that the model with more neurons has a better information representation ability and a better training effect. Initially, testing NRMSE was greater than training NRMSE; this phenomenon is called overfitting. As the reservoir size reduced, the training error went down, and the overfitting was improved, confirming that reducing the network size is a directly effective approach to improving the generalization performance of the network. There is no doubt that pruning neurons will lead to a reduction in network size, which is the reason why the unpruned DeepESN is tested as a benchmark, and a successful pruning algorithm should outperform this benchmark. Furthermore, the minimum testing NRMSE condition is chosen to compare the extreme generalization performance of each experiment group, which was recorded in Table 1.

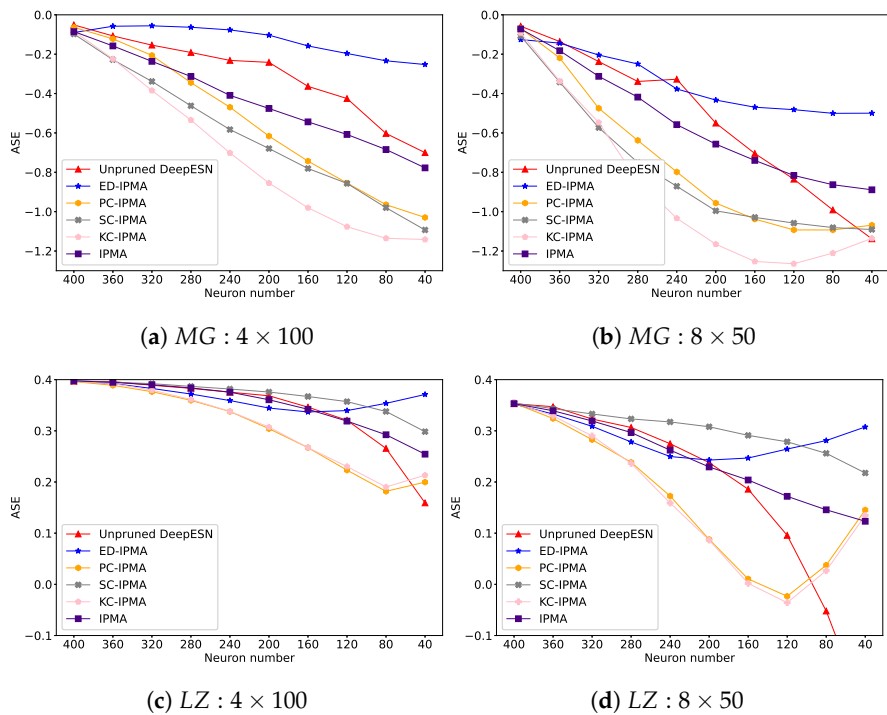

(**a**) $MG : 4 \times 100$

(**b**) $MG : 8 \times 50$

(**c**) $LZ : 4 \times 100$

(**d**) $LZ : 8 \times 50$

**Figure 8.** ASE comparison of unpruned DeepESN and DeepESN pruned by IPMA, ED-IPMA, PC-IPMA, SC-IPMA, KC-IPMA. (**a**): Initial 4 layer reservoirs with 100 neurons in each reservoir on *MG* dataset; (**b**): Initial 8 layer reservoirs with 50 neurons in each reservoir on *MG* dataset; (**c**): Initial 4 layer reservoirs with 100 neurons in each reservoir on *LZ* dataset; (**d**): Initial 8 layer reservoirs with 50 neurons in each reservoir on *LZ* dataset.

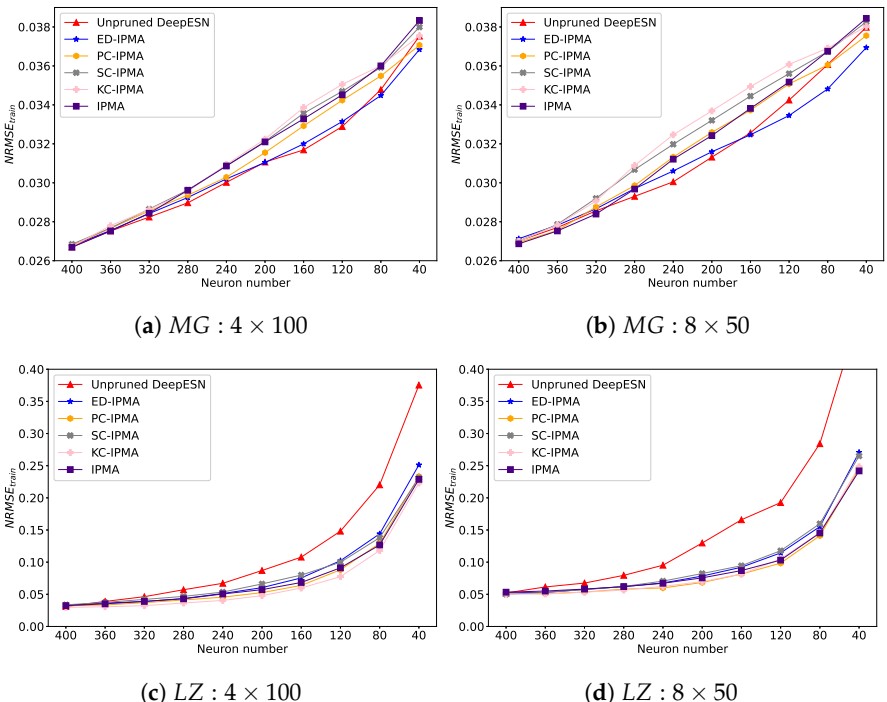

(**a**) $MG : 4 \times 100$

(**b**) $MG : 8 \times 50$

(**c**) $LZ : 4 \times 100$

(**d**) $LZ : 8 \times 50$

**Figure 9.** Training NRMSE comparison of unpruned DeepESN and DeepESN pruned by IPMA, ED-IPMA, PC-IPMA, SC-IPMA, KC-IPMA. (**a**): Initial 4 layer reservoirs with 100 neurons in each reservoir on *MG* dataset; (**b**): Initial 8 layer reservoirs with 50 neurons in each reservoir on *MG* dataset; (**c**): Initial 4 layer reservoirs with 100 neurons in each reservoir on *LZ* dataset; (**d**): Initial 8 layer reservoirs with 50 neurons in each reservoir on *LZ* dataset.

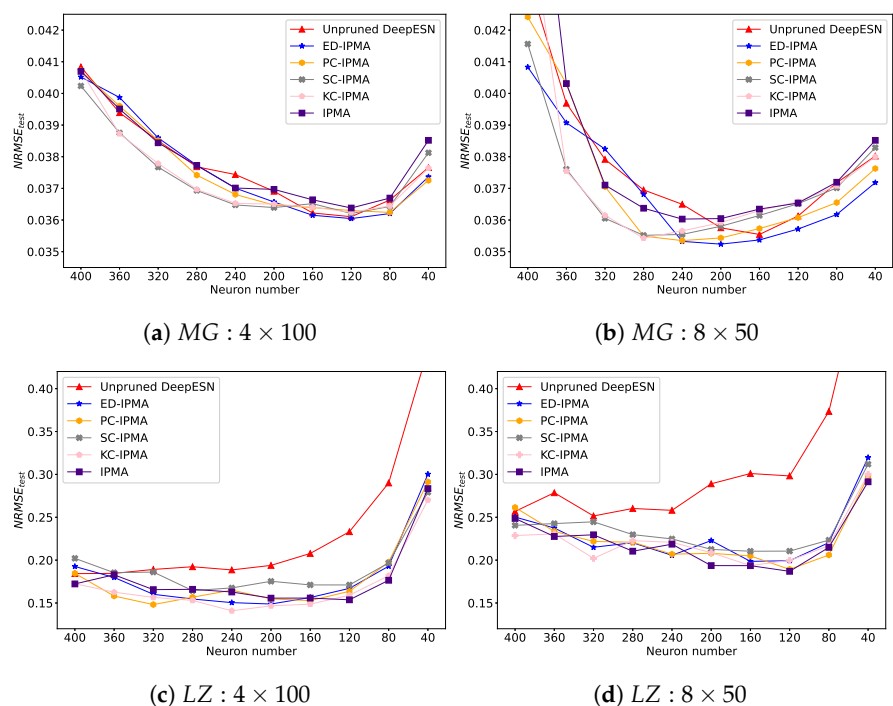

<div align="center">(<b>a</b>) <i>MG</i> : 4 × 100   (<b>b</b>) <i>MG</i> : 8 × 50</div>

<div align="center">(<b>c</b>) <i>LZ</i> : 4 × 100   (<b>d</b>) <i>LZ</i> : 8 × 50</div>

**Figure 10.** Testing NRMSE comparison of unpruned DeepESN and DeepESN pruned by IPMA, ED-IPMA, PC-IPMA, SC-IPMA, KC-IPMA. (**a**): Initial 4 layer reservoirs with 100 neurons in each reservoir on *MG* dataset; (**b**): Initial 8 layer reservoirs with 50 neurons in each reservoir on *MG* dataset; (**c**): Initial 4 layer reservoirs with 100 neurons in each reservoir on *LZ* dataset; (**d**): Initial 8 layer reservoirs with 50 neurons in each reservoir on *LZ* dataset.

Similarly, from Figure 8, the DeepESN pruned by ED-IPMA maintained a minimum ASE loss as the number of neurons decreases, and we suspect that there were some hidden relationships between the Euler distance and Renyi's quadratic entropy. From Table 1, the unpruned DeepESNs had a greater standard deviation of the minimum testing NRMSE compared to the pruned DeepESN, indicating that the NS-IPMA method has good robustness.

On the MG dataset: From Figure 10a,b, in the early stage, the testing error of DeepESN pruned by SC-IPMA and KC-IPMA dropped rapidly; later, DeepESN pruned by ED-IPNA and PC-IPMA performed best when the majority of neurons had been pruned. From Table 1, although ED-IPMA achieved the best extreme generalization performance, the pruned DeepESNs had no obvious improvement in the mean value of minimum testing NRMSE compared to the unpruned DeepESN.

On the LZ dataset: From Figures 9c,d, 10c,d and Table 1, the training error, testing error, and extreme generalization performance of pruned DeepESNs were all significantly improved compared to unpruned DeepESN, and the diversity of different similarity estimation criteria were not prominent. The noncriterion-based IPMA method performed as well as criterion-based NS-IPMA methods.

In summary, all these experimental results showed that the NS-IPMA method is a successful approach to improving the generalization performance of DeepESN, which is specific, in almost all of our designed experiments, in that the DeepESN pruned by NS-IPMA has better generalization performance than the standard unpruned DeepESN.

**Table 1.** Minimum testing NRMSE condition of each experiment group.

| Group | | Unpruned DeepESN | Pruned DeepESN | | | | |
|---|---|---|---|---|---|---|---|
| Method | | - | ED-IPMA | PC-IPMA | SC-IPMA | KC-IPMA | IPMA |
| $MG : 4 \times 100$ | $NRMSE_{test}$ (Mean) | 0.036 105 | **0.036 046** | 0.036 240 | 0.036 206 | 0.036 243 | 0.036 380 |
| | $NRMSE_{test}$ (Std.) | 0.000 483 | 0.000 388 | 0.000 221 | 0.000 281 | 0.000 144 | 0.000 291 |
| | Neuron number | 120 | 120 | 80 | 120 | 120 | 120 |
| $MG : 8 \times 50$ | $NRMSE_{test}$ (Mean) | 0.035 544 | **0.035 238** | 0.035 350 | 0.035 516 | 0.035 427 | 0.036 030 |
| | $NRMSE_{test}$ (Std.) | 0.000 699 | 0.000 346 | 0.000 516 | 0.000 533 | 0.000737 | 0.000 338 |
| | Neuron number | 160 | 200 | 240 | 280 | 280 | 240 |
| $LZ : 4 \times 100$ | $NRMSE_{test}$ (Mean) | 0.184 285 | 0.148 812 | 0.152 775 | 0.164 455 | **0.140 930** | 0.155 667 |
| | $NRMSE_{test}$ (Std.) | 0.043 201 | 0.026 072 | 0.024 417 | 0.022 953 | 0.031 862 | 0.027 816 |
| | Neuron number | 400 | 200 | 160 | 280 | 240 | 200 |
| $LZ : 8 \times 50$ | $NRMSE_{test}$ (Mean) | 0.251 535 | 0.198 581 | 0.189 230 | 0.210 376 | 0.193 678 | **0.186 933** |
| | $NRMSE_{test}$ (Std.) | 0.044 484 | 0.027 631 | 0.024 995 | 0.033 404 | 0.029 164 | 0.024 092 |
| | Neuron number | 320 | 160 | 120 | 160 | 160 | 120 |

Red bold values indicate the best validation performance of all methods.

## 6. Conclusions and Prospects

In our research, a new iterative pruning merging algorithm was proposed to simplify the architecture of DeepESN. As to which neurons should be pruned out, four different similarity estimation criteria were attempted. The unpruned DeepESNs is a benchmark that identifies the evolutionary characteristic of network performance by reducing network size, and the effectiveness of the proposed method was experimentally verified by comparing pruned DeepESNs with unpruned DeepESNs in the same network size. The results showed that these NS-IPMA methods have good network structure adaptability, and the DeepESNs pruned by the NS-IPMA method have better generalization performance and better robustness than unpruned DeepESNs, indicating that the NS-IPMA method is a feasible and superior approach to improving the generalization performance of DeepESN. The NS-IPMA method provides a novel approach for choosing the appropriate network size of DeepESN. One could start with a larger model than necessary for reservoir size and then prune or merge some similar neurons to obtain a better DeepESN model. One could select a simple architecture with small computation requirements while keeping the testing error acceptable.

In many tasks, low computational cost and reliable performance cannot be achieved at the same time. The experimental results showed that the newly proposed method could better balance the computational cost and performance; therefore, it has broad application prospects in real-time (RT) systems, including RT control, RT forecast, RT decision, etc., for example, RT control for modernized microgrids [29], intelligent Internet of Things automatic irrigation control system [30], lower generation data prediction for wind power plant [31], trading prediction and dynamic decisions for online trading systems [32], model predictive control [33,34], predictive feedforward control for servo systems [35,36].

However, there are still some shortcomings in our work. First, the problem of how to choose the redundant neurons to be pruned out or what the best neuronal similarity estimation criterion should be, remains unsolved. Second, only the hierarchical structure, the reservoir diversity, and the overall error performance are investigated, and more evolutionary characteristics of different reservoirs, such as their spectral radius, resulting from the NS-IPMA method, are not analyzed. Third, the effects of pruning and merging are not clearly distinguished. Future research on the NS-IPMA method could focus on the theoretical analysis of the above-unsolved problem or on practical applications where high-performance computing is required.

**Author Contributions:** Methodology, Q.S.; software, Q.S.; validation, Q.S.; investigation, Q.S.; resources, Q.S.; writing—original draft, Q.S.; writing—review and editing, Q.S. and H.Z.; supervision, Y.M.; funding acquisition, Y.M. All authors have read and agreed to the published version of the manuscript.

**Funding:** This research was funded by the National Natural Science Foundation of China—grant number: 62271109.

**Institutional Review Board Statement:** Not applicable.

**Informed Consent Statement:** Not applicable.

**Data Availability Statement:** The data presented in this research and relevant resources are available upon request; please send your request message to shen99855@outlook.com.

**Conflicts of Interest:** The authors declare no conflict of interest.

**Abbreviations**

The following abbreviations are used in this manuscript:

| | |
|---|---|
| RNN | Recurrent neural network |
| ESN | Echo state network |
| LSTM | Long short-term memory |
| GRU | Gated recurrent unit |
| DNN | Deep neural network |
| DeepESN | Deep echo state network |
| SIPA | Sensitive iterative pruning algorithm |
| SCRN | Simple cycle reservoir network |
| NS-IPMA | Neuronal similarity-based iterative pruning merging algorithm |
| LI-ESN | Leaky integrator echo state network |
| ESP | Echo state property |
| RC | Reservoir computing |
| ASE | Average state entropy |
| ED | Euclidean distance |
| PC | Pearson's correlation |
| SC | Spearman's correlation |
| KC | Kendall's correlation |
| ED-IPMA | NS-IPMA based on the inverse of Euclidean distance criterion |
| PC-IPMA | NS-IPMA based on the inverse of Pearson's correlation criterion |
| SC-IPMA | NS-IPMA based on the inverse of Spearman's correlation criterion |
| KC-IPMA | NS-IPMA based on the inverse of Kendall's correlation criterion |
| IPMA | Iterative pruning merging algorithm |
| PSO | Particle swarm optimization |
| *MG* | Mackey–Glass |
| *LZ* | Lorenz z-axis |
| DC | Direct component |
| NRMSE | Normalized root mean square error |
| RT | Real-time |

**Appendix A. Hyperparameter Tuning**

As defined in Section 2.2, hyperparameters ($\alpha$, $\gamma_i$, $\gamma_r$, $\gamma_p$ and $\lambda$) play essential roles in the performance of DeepESN, as well as in the successful application of NS-IPMA. $\alpha$ and $\gamma_r$ affect the stability of reservoirs, $\alpha = 0.92$ and $\gamma_r = 0.8$ are set to satisfy Equation (6); $\lambda$ affects the generalization performance, a small regularization factor $\lambda = 1 \times 10^{-10}$ is chosen to make the output weights better fit the training samples. $\gamma_i$ adjusts the strength of the input signal into the first layer of DeepESN, $\gamma_p$ adjusts the strength of the input signal into higher layers of DeepESN; thus, $\gamma_i$ is optimized on the LI-ESN, which is the first layer of DeepESN. After that, $\gamma_p$ is optimized on the DeepESN after higher layers are hierarchically stacked on the original LI-ESN. The tuned results are recorded in Table A1.

**Table A1.** Hyperparameters applied under different initial reservoir sizes on different datasets.

| Dataset | $MG$ | $MG$ | $LZ$ | $LZ$ |
| Initial Size | $4 \times 100$ | $8 \times 50$ | $4 \times 100$ | $8 \times 50$ |
| --- | --- | --- | --- | --- |
| $\alpha$ | 0.92 | 0.92 | 0.92 | 0.92 |
| $\gamma_r$ | 0.8 | 0.8 | 0.8 | 0.8 |
| $\gamma_i{}^{*1}$ | 0.373 84 | 0.253 14 | 0.096 31 | 0.064 94 |
| $\gamma_p{}^{*2}$ | 0.211 36 | 0.241 62 | 0.335 51 | 0.347 11 |
| $\lambda$ | $1 \times 10^{-10}$ | $1 \times 10^{-10}$ | $1 \times 10^{-10}$ | $1 \times 10^{-10}$ |

*1 Tuned by the PSO in the range of $[1 \times 10^{-5}, 10]$; *2 tuned by PSO in the range of $[0.1, 5]$.

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
