# Peer review of "Improving Deep Echo State Network with Neuronal Similarity-Based Iterative Pruning Merging Algorithm"

_applsci, doi:10.3390/app13052918_

Round 1

Reviewer 1 Report

 It Seems to be inconsistent:

1)"During pruning, the number of remaining neurons in each reservoir of

pruned DeepESN, which was processed by different similarity estimation cri-terion based NS-IPMA, are shown in Fig. 5 ¨ (page 15)

Author Response

Dear reviewer:

We greatly appreciate your feedback and suggestions. Based on the feedback, we have carefully amended the manuscript to improve the quality of this paper.

Besides, the manuscript has been double-checked, and the typos and grammatical errors have been corrected in the revised manuscript. All changes in the revised manuscript have been highlighted in yellow. We plan to upload the revised manuscript on Feb.14th.

Here are the point-by-point responses to the comments:

Comment 1: It Seems to be inconsistent:” During pruning, the number of remaining neurons in each reservoir of pruned DeepESN, which was processed by different similarity estimation criterion based NS-IPMA, are shown in Fig. 5”  (page 15)

Response: Many thanks for your reminder, This sentence is modified to: “The number of neurons remaining during pruning in each reservoir of DeepESN, which was pruned by different similarity estimation criterion-based NS-IPMA methods, are shown in Figure 7”.

Reviewer 2 Report

Dear Authors,

I have some comments on your article:

1. The article should be formatted according to the template required by the Applied Sciences journal.

2. Please check all equations, symbols, and indexes used in the text and equations.

3. 4.1. Datasets - The description should be greatly expanded.

4. Literature should be checked if there are no newer items. Especially from the last 18 months. It would be good to add several references.

5. In the summary, more should be written about the practical application of the proposed algorithm.

Author Response

Dear reviewer:

Best wishes!

We greatly appreciate your feedback and suggestions. Based on the feedback, we have carefully amended the manuscript to improve the quality of this paper.

Besides, the manuscript has been double-checked, and the typos and grammatical errors have been corrected in the revised manuscript. All changes in the revised manuscript have been highlighted in yellow. We plan to upload the revised manuscript on Feb.14th.

Please download the attachment to see point-by-point responses.

Reviewer 3 Report

The authors' goal in this study is to address the issue of selecting the DeepESN's optimal size. A neuronal similarity-based iterative pruning merging method (NS-IPMA), which is inspired by the sensitive iterative pruning technique, is suggested to iteratively prune or merge the most comparable neurons in Deep-ESN. To illustrate the utility of NS-IPMA, two chaotic time series prediction tasks are used. The results demonstrate that the NS-IPMA-pruned DeepESN beats the unpruned DeepESN with the same network size and that NS-IPMA is a practical and effective method for enhancing DeepESN's generalization performance. The paper generally sounds good, but needs some revisions:

- Add a section of main results to better clarify what is new in this paper

-providing codes will be helpful for readers

-add a flowchart and pseudo codes for your pruning algorithm

-statistical analysis is required to have a fair comparison with other structures, and show the effectiveness of the proposed scheme

-add a remark on how your approach can be used in control systems such as: Optimal deep learning control for modernized microgrids

-add some direction for future studies

Author Response

(The authors gave the same response as above.)

Round 2

Reviewer 2 Report

Dear Authors,

Thank you very much for introducing changes that have improved the quality of the article. I have no more comments.

Reviewer 3 Report

the paper can be accepted in My opinion